The interaction of InvF-RNAP is mediated by the chaperone SicA in Salmonella sp: an in silico prediction

Farias André B. 1 2 bfarias.andre@gmail.com
Cortés-Avalos Daniel 3
Ibarra J. Antonio 3
http://orcid.org/0000-0002-6879-0673 Perez-Rueda Ernesto 1
1 Instituto de Investigaciones en Matemáticas Aplicadas y en Sistemas, Unidad Académica del Estado de Yucatán, Universidad Nacional Autónoma de México , Mérida, Yucatán , Mexico
2 Laboratório Nacional de Computação Científica—LNCC , Petrópolis, Rio de Janeiro , Brazil
3 Laboratorio de Genética Microbiana, Departamento de Microbiología, Escuela Nacional de Ciencias Biológicas, Instituto Politécnico Nacional, Universidad Nacional Autónoma de México , Ciudad de México, Ciudad de México , México
Ettrich Rudi
Electronic publication date: 2024 Mar 25
Publication date: 2024
Volume: 12
Electronic Location ID: e17069
Received 2023 Oct 30; Accepted 2024 Feb 18
Copyright: © 2024 Farias et al.
Copyright year: 2024
Copyright holder: Farias et al.
License: This is an open access article distributed under the terms of the Creative Commons Attribution License, which permits unrestricted use, distribution, reproduction and adaptation in any medium and for any purpose provided that it is properly attributed. For attribution, the original author(s), title, publication source (PeerJ) and either DOI or URL of the article must be cited.
License URL: https://creativecommons.org/licenses/by/4.0/

Keywords: InvF-SicA complex, Molecular dynamics simulations, Salmonella invasion chaperone, Alanine scanning mutagenesis

Funding: PAPIIT-DGAPA UNAM grant IN220523 (E.P-R) Coordenação de Aperfeiçoamento de Pessoal de Nível Superior CAPES/UNAM, COOPBRAS, n° 05/2019, 88887.368759/2019-00—Finance Code 001 CONAHCYT A1-S-25438 and SIP-IPN 2024-0122 PhD fellowship (235269) from CONAHCYT This work was supported by PAPIIT-DGAPA UNAM grant IN220523 (E.P-R), and by the Coordenação de Aperfeiçoamento de Pessoal de Nível Superior - CAPES/UNAM, COOPBRAS, n° 05/2019, 88887.368759/2019-00 - Finance Code 001. The research was supported by projects CONAHCYT A1-S-25438 and SIP-IPN 2024-0122 (to José A Ibarra). Daniel Cortés-Avalos received support in the form of a PhD fellowship (235269) from CONAHCYT. The funders had no role in study design, data collection and analysis, decision to publish, or preparation of the manuscript.

==============================
In this work we carried out an in silico analysis to understand the interaction between InvF-SicA and RNAP in the bacterium Salmonella Typhimurium strain LT2. Structural analysis of InvF allowed the identification of three possible potential cavities for interaction with SicA. This interaction could occur with the structural motif known as tetratricopeptide repeat (TPR) 1 and 2 in the two cavities located in the interface of the InvF and α-CTD of RNAP. Indeed, molecular dynamics simulations showed that SicA stabilizes the Helix-turn-Helix DNA-binding motifs, i.e., maintaining their proper conformation, mainly in the DNA Binding Domain (DBD). Finally, to evaluate the role of amino acids that contribute to protein-protein affinity, an alanine scanning mutagenesis approach, indicated that R177 and R181, located in the DBD motif, caused the greatest changes in binding affinity with α-CTD, suggesting a central role in the stabilization of the complex. However, it seems that the N-terminal region also plays a key role in the protein-protein interaction, especially the amino acid R40, since we observed conformational flexibility in this region allowing it to interact with interface residues. We consider that this analysis opens the possibility to validate experimentally the amino acids involved in protein-protein interactions and explore other regulatory complexes where chaperones are involved.

Introduction

Salmonella species are Gram-negative, flagellated, facultatively anaerobic bacteria, widely distributed around the world (Mellor et al., 2019). The genus Salmonella contains two species, according to the Centers for Disease Control and Prevention (CDC) system, (i) enterica, divided in six subspecies (enterica, salamae, arizonae, diarizonae, houtenae, and indica), and (ii) bongori, formerly called subspecies V (Brenner et al., 2000). S. enterica presents a huge diversity of species, with more than 2,600 unique serovars (Cheng, Eade & Wiedmann, 2019), being most associated with human diseases (Gal-Mor, Boyle & Grassl, 2014). Therefore, there is a global concern, since the CDC estimation of infections is around 1.35 million, 26,500 hospitalizations, and 420 deaths in the United States every year. Moreover, Salmonella spp. is one of the major causes of diarrheal disease throughout the world and foodborne pathogens (Mukherjee et al., 2019). In this regard, bacteria have developed complex gene transcription mechanisms in response to environmental change, allowing them to survive and adapt to their environment over the years (Browning & Busby, 2016). Despite new therapies have been developed due to the increased molecular knowledge about essential bacterial genes and their role in mechanism regulation, there is still a lack of information about structural mechanism.

The gene expression has a crucial step the recognition of the promoter and the initiation of transcription by RNA polymerase (RNAP) (Browning & Busby, 2004). The role of chaperones in the regulation of type III secretion systems (T3SS) has been described for many years (Darwin, 2001; Feldman & Cornelis, 2003). SicA, a specific chaperone for SipB and SipC of Salmonella spp, acts together with InvF to activate the expression of essential genes for Salmonella pathogenicity. The complex InvF/SicA regulates the expression of sicA, sopB, sptP, sopE, sopE2, STM1239, and several other genes. Preliminary studies suggest that SicA acts as a cofactor to InvF transcription (Darwin & Miller, 2000), although it is not yet clear how SicA interacts with InvF, RNAP, DNA, or in complex with them. The hypothesis is that SicA should act as a cofactor for InvF, once they could be co-purified and a direct interaction observed (Darwin, 2001; Kim et al., 2014). A recent study shows that transcriptional activation of sopB requires both InvF and SicA, suggesting the role of a complex between InvF and SicA as a transcriptional regulator to induce the expression of virulence genes (Romero-González et al., 2020). InvF has two structural domains, (i) the amino terminus, where resides the signal-binding or dimerization domain and the (ii) carboxy terminus, where the DNA-binding domain (Ibarra et al., 2008) has been identified. The interaction site with the RNAP is likely located in the C-terminus, as identified in other AraC/XylS-like family members such as PerA (Lara-Ochoa et al., 2021), MelR (Grainger et al., 2004), and XylS (González-Pérez et al., 2002).

In contrast, the SicA structure is composed of eight alpha-helical regions (H1–H8) and three tetratricopeptide repeat (TPR) motifs (H2/H3, H4/H5, and H6/H7). The N and C-terminus has another two helices, H1, and H2, respectively (Pallen, Francis & Fütterer, 2003). These TPR motifs play an important role in protein-protein interaction, being the key to transcriptional function of the SicA chaperone (Kim et al., 2014). In order to understand how SicA interacts with InvF and RNAP, an exhaustive in silico analysis was carried out to identify pockets and protein-protein interactions. Structural modeling, molecular dynamics and AA mutagenesis, suggest that InvF is stabilized by SicA, mainly in the region of DNA-Binding Domain (DBD), and such stabilization is important for InvF/SicA-mediated gene activation.

Methods

Structural model prediction and protein preparation

The 249 amino acids sequence of invasion protein InvF (ID: A0A0F7JCN6) from S. Typhimurium strain LT2 was obtained from the Uniprot database (The UniProt Consortium, 2023). From this sequence, the Robetta (Baek et al., 2021) program with default parameters, was used for structural model prediction. The amino acid sequence of SicA from S. Typhimurium strain LT2 was obtained from Uniprot (ID: P69066) and used to find templates on the SWISS-MODEL (Waterhouse et al., 2018) server. The chaperone protein IpgC from S. flexneri (PDB code: 3GZ2), which had the highest score of Sequence Coverage, Identity, Global Model Quality Estimation (GMQE) and Quaternary Structure Quality Estimate (QSQE), was chosen to build the 3D-structure of monomeric SicA, by homology modeling. The protein quality scores (z-score) was evaluated in ProSA web server (Wiederstein & Sippl, 2007) and geometric parameters of structures were validated with PROCHECK (Laskowski et al., 1993) from the UCLA-DOE LAB (saves.mbi.ucla.edu) server, which generated the Ramachandran plot (see Fig. S1 and Table S1).

The structure of carboxy-terminal domain of the alpha subunit of RNAP ( α-CTD) was obtained by solution NMR, in which it was complexed with the structure of MarA and DNA (PDB ID 1XS9) (Dangi et al., 2004). Considering that the structure of MarA, complexed with the α-CTD and DNA, is similar to the structure of InvF (RMSD of 2.07 Å), and belongs to the same family, AraC/XylS family, a structural superposition was performed to obtain the α-CTD coordinates in relation to InvF. The system was minimized with 50,000 steps of steepest descent algorithm to avoid any atoms overlapping (Fig. S2).

Binding pocket detection

In order to identify a binding site where SicA could interact with the InvF- α-CTD complex, a detection cavity strategy was performed. The structure of the InvF- α-CTD complex was submitted in CavityPlus (Xu et al., 2018; Yuan, Pei & Lai, 2013), a web server package to detect cavities in a protein structure, that ranks them based on ligandability and druggability scores. This approach is a geometry-based method that employs a probe sphere. The cavities detected by this approach provide information about the shapes, druggability of a pocket and a prediction of average pKd. The cavities predicted were analyzed in PyMOL software (v. 2.4.0a0; Schrodinger, 2010).

Construction of protein complexes by molecular docking

The interaction mode of SicA with the InvF- α-CTD complex was obtained through molecular docking using the HADDOCK server (van Zundert et al., 2016). The residues of the three best scored cavities (Cav1, Cav2 and Cav3) of InvF/ α-CTD complex were used as active residues to molecular docking. The amino acids from SicA structure defined as active residues were Q34-D66, P71-L100, and P107-R134, referring to the tetratricopeptide repeat motifs, TPR1, TPR2 and TPR3, respectively. No passive residues were defined and all docking parameters were set to default.

Molecular dynamics simulation

Molecular dynamics simulations were performed with the Gromacs (Van Der Spoel et al., 2005) program (v. 2020.2) with the force field GROMOS 54a7 (Schmid et al., 2011), and water model SPC (Berendsen et al., 1981). Simulations were performed under periodic boundary conditions based on cubic boxes with 1 nm of distance between the solute and the box. The system was neutralized with Na+ or Cl− counterions to attain equilibration. The motion equation was integrated using a leapfrog scheme with a time step of 2 fs. All bonds involving hydrogen atoms were constrained using the LINCS (Hess et al., 1997) procedure. A cutoff distance of 1.2 nm was applied for the simple truncation of the Lennard-Jones interaction, and to define the real-space region of the electrostatic interactions. The Smooth Particle Mesh Ewald (SPME) (Essmann et al., 1995) method was used with a grid spacing of 0.12 nm. Pair lists were generated and updated according to the Verlet schema (Páll & Hess, 2013). Therefore, the systems considered were energy minimized with 50,000 steps of the steepest descent algorithm, and subsequently equilibrated. Equilibration of the system was performed using positional restraints on the protein atoms with gradual heating, incrementing 50 K, from 50 to 300 K, over 1.2 ns. The simulations were performed on the NPT set with a reference pressure of 1 bar, and a reference temperature of 300 K. The temperature was kept constant by a velocity-rescaling algorithm (Bussi, Donadio & Parrinello, 2007) with the protein and solvent degrees of freedom coupled separately to temperature baths, with a coupling constant of 0.1 ps. The pressure was kept close to the reference pressure P by isotropic coupling using the Parrinello-Rahman algorithm (Parrinello & Rahman, 1980). All simulations were performed in duplicated with 500 ns of trajectory each.

Binding affinity calculations by MM-PBSA

The binding affinity of proteins was assessed by Molecular Mechanics Poisson-Boltzmann Surface Area (MM-PBSA) using the package g_mmpbsa (Kumari et al., 2014). The general expression of free energy for binding of proteinA to the proteinB to form the complex can be written as

(1) ΔGbinding=Gcomplex−(GproteinA+GproteinB)

where GproteinA is the free energy for the binding of protein A to protein B ( GproteinB) forming a protein complex ( Gcomplex). The above equation can be rewritten in terms of its enthalpic ( ΔH), and entropic ( TΔS) contributions.

(2) ΔGbinding=ΔH−TΔS=ΔEMM+ΔGsol−TΔS

where ΔEMM is the average potential energy of molecular mechanics in vacuum (bonded and non bonded terms), and ΔGsol means the free energy of solvation (polar and non polar terms). ΔEMM and ΔGsol can be rewritten as

(3) ΔEMM=ΔEbond+(ΔEvdW+ΔEelec)ΔGsol=ΔGpolar+ΔGnonpolar.

Changes in internal energy (e.g., angles, bonds and dihedrals) are represented by the bonded term ( Ebond), while van der Waals ( EvdW) and electrostatic ( Eelec) energies represent the unbonded terms ( Enonbond). Solvation free energy ( ΔGsol) was calculated using the Poisson-Boltzmann model for polar contribution ( ΔGpolar), while nonpolar contribution ( ΔGnonpolar) was calculated with SASA. The entropic contributions are usually neglected (Wang et al., 2019) due heavy computational cost, time-consuming and the magnitude of standard error is high compared to the other energetic terms. In this way, the equation can be expressed as follows

(4) ΔGbinding=ΔEbond+(ΔEvdW+ΔEele)+ΔGpolar+ΔGnonpolar−TΔS.

In the present work, for each simulations, five windows of 2 ns (200 snapshots) were extracted from trajectory of molecular dynamics simulations. A total of 1,000 snapshots, for each simulation, were subjected to MM-PBSA calculations. First, we assessed the binding affinity between SicA (interacting via TPR1, TPR2 and TPR3) and InvF/ α-CTD complex. After that, we calculated the binding energy between α-CTD and InvF.

Computational alanine scanning mutagenesis (ASM)

In order to evaluate the effect of InvF residue mutation in the molecular stability and binding affinity of the InvF/ α-CTD complex, the first frame of the same windows used in the PBSA calculations previously described, were used to perform systematic amino acid mutations with PyMOL (Schrodinger, 2010). Based on the energy contribution of amino acids, an energy threshold of -50 kJ/mol was applied to select the amino acids to be mutated. Thus, amino acids M1, K13, R14, K15, R17, R40, K99, K134, R140, K141, R176, R177 and R181 related to InvF were mutated.

After residue mutation, the topology of the mutated protein was generated with the Gromacs (Van Der Spoel et al., 2005), and the systems were minimized, equilibrated as previously described. All simulations were performed using five windows of 2 ns (1,000 frames) of trajectory. The software g_mmpbsa (Kumari et al., 2014) was used to calculate the binding free energy of the mutated protein, obtaining the difference ( ΔΔGbinding) upon alanine mutation.

(5) ΔΔGbinding=ΔGbindingmut−ΔGbindingwt.

Results and discussion

Characterization of protein-protein interactions in InvF- α-CTD complex

Firstly, a model of the InvF protein from S. Typhimurium was built based on its amino acid sequence obtained from the Uniprot database (ID A0A0F7JCN6). The structural prediction of the InvF model obtained by the Robetta program was validated according to the geometric parameters, as can be seen in the Ramachandran graph (Fig. S1 and Table S1). Ramachandran analysis indicated that 93.3%, 4.4% and 0.9% of the residues were located in favored, allowed regions and disallowed regions, respectively. In addition, our model shows −7.12 (z-score) indicating the good quality of our model.

In order to have a structural model of the InvF/ α-CTD complex, a structural alignment between InvF and the crystal of MarA (PDB ID 1XS9 (Dangi et al., 2004)), which is in complex with the structure of α-CTD of the RNAP, and DNA, was performed. Figure 1A presents the structure of the InvF. The structural alignment between the InvF model and MarA crystal (Fig. 1B), indicated a good overlap between both structures, with a Root Mean Square Deviation (RMSD) value of 2.07 Å over 649 atoms. This structural alignment allowed us to obtain a possible position of α-CTD referring to the structure of InvF, in which some interactions between amino acids are conserved (Figs. 1C and 1D).

Figure 1 Cartoon representation of InvF-model protein (A). Overlap of the InvF model (gray) to MarA structure (light green) and α-CTD (orange) (B). Comparison between amino acids interactions, within the 3Å of distance, of protein-protein interface of MarA (C) and InvF (D) are represented by line and yellow dashed lines.

Previous works have reported the importance of certain MarA amino acids for RNAP affinity. For instance, Gillette, Martin & Rosner (2000) showed that the DNA-binding activity of the MarA mutant W19A was severely affected (promoter affinity), corroborating the model proposed by Dangi et al. (2004). In addition, it was observed that W19F and W19Y are partially functional with activities ranging from 37% to 97% of wild-type (Gillette, Martin & Rosner, 2000). We noticed a Y150 in the InvF model in the equivalent position of W19, and it is directed to R265, reported as an important RNAP amino acid responsible for affinity (Dangi et al., 2004) (Figs. 1D and S4). Interactions of amino acids D18, D22, S37 with α-CTD through their backbone atoms of N268, R265 and N294, were also described in the MarA model, although the alanine mutation at these positions had minimal effects in the promoter affinity (Gillette, Martin & Rosner, 2000). From the alignment of InvF structure, it was also possible to observe that amino acids R140, Y150 and D167 were able to interact with these residues of α-CTD. These results suggest that the strategy of aligning the InvF structure to the MarA model, aiming to obtain the α-CTD coordinates in relation to InvF produced similar protein-protein interaction to a stable complex.

Molecular dynamics (MD) simulations were carried out to assess the stability α-CTD coordinates in relation to InvF. Thus, a simulation revealed that, in fact, α-CTD remains stable and interacting with InvF throughout the 500 ns trajectory. The analysis of the distance between the center of mass of the α-CTD and the center of mass of the InvF shows that there was no significant deviation (Fig. 2A). Although the structure of the α-CTD remains stable, it can be seen that some InvF amino acids present in the N-terminal domain (NTD) have enough flexibility to perform a movement that promotes interactions with amino acids of the α-CTD (Fig. 2B). This structural change of the complex can also be observed by the RMSD plot (blue line 60 ns), and should represent an affinity increase in relation to the initial orientation. MM-PBSA calculations revealed that the binding energy of α-CTD to InvF is −752.82 ± 103.30 kJ/mol, pointing to the role of some amino acids for protein-protein interactions (Fig. 2C).

Figure 2 Molecular dynamics simulations of InvF and α-CTD along 500 ns of trajectory for two replicas (rep1 and rep2).

Distance of the mass center of α-CTD in relation to InvF (black and gray line) and the RMSD of InvF (blue and purple line) and α-CTD (brown and orange line) (A). Comparison between first and last frame of MD simulations (B). Energy contribution of InvF (C) and α-CTD (D) residues calculated by MM-PBSA.

MM-PBSA calculations showed that R176 and R181, located in the DBD motif, present higher values of binding energy suggesting a critical role in protein-protein affinity or in structural stabilization by intramolecular interactions. In addition, protein-protein interactions are also occurring with some residues from NTD, including K13, R14, K15 and R40. These residues, represented as sticks in Fig. 2B, have the capability of interacting with α-CTD, mainly with Y277, D280, I326 and E329. The energy profile of α-CTD amino acids is illustrated in Fig. 2D. Our results revealing a hypothetical mechanism of α-CTD recognition by InvF, pointing important residues from the N-terminal domain to binding affinity. To note, InvF requires a chaperone protein SicA to form a complex and activate its genes (Darwin & Miller, 2000; Romero-González et al., 2020). However, the mechanism underlying interactions between InvF and SicA and how this complex is able to activate the transcription remains unclear. In order to evaluate the role of SicA in molecular stability of InvF, a systematic study was carried out to find cavities and modes of interaction.

Searching for cavities in the structure of InvF/ α-CTD complex

Although the results show that InvF has sufficient affinity with α-CTD, recent work has shown the need for the chaperone SicA to act as a co-activator to functionalize InvF (Romero-González et al., 2020). It is known that SicA interacts with InvF, but the role of SicA in stimulating the InvF activator is not entirely understood. In this way, we performed a study of the pockets present on the surface of the InvF/ α-CTD complex in order to identify regions of interaction with SicA. Figure 3 presents the three best classified cavities according to the CavityPlus program (all the results are included in Table S2). Cavities one and three are located in an interface region of InvF with the α-CTD, while Cavity two is closer to the NTD region. In addition, the three cavities predicted are not located in the region of contact with DNA, which is between the α10 and α7, suggesting that SicA interaction in these cavities would not interfere with protein-DNA interaction.

Figure 3 The three best cavities found in the InvF/ α-CTD complex.

InvF and α-CTD is colored in gray and orange, respectively.

After determining possible interaction regions, we evaluated the interaction of SicA with the InvF/ α-CTD complex by molecular docking. Nonetheless, SicA has three tetratricopeptide repeat motifs (TPR) and it is still unknown which specific conformation of SicA is responsible for binding affinity. Mutation of TPR motifs of SicA greatly affects the transcription of sigDE and sopE, indicating the relevance of TPR motif for the transcriptional cofactor function (Kim et al., 2014). Thus, a systematic study of docking was carried out in order to evaluate the interaction of the three TPRs motifs against the three cavities predicted. Although the scoring functions obtained through molecular docking calculations do not correlate with binding affinity (Kastritis & Bonvin, 2010), this value can be used to compare different solutions. Figure S5 presents the score values, the buried surface area (BSA), and the energy contributions (E_ele + E_vdw) of the docking solutions (all solutions are shown in Tables S3–S5).

The docking results revealed that SicA preferentially interacts using TPR1 and TPR2 in cavities 3 and 1, respectively. Our results are in agreement with the model proposed by Li et al. (2022) in which c-di-GMP interacts with SicA through residues T25, K27, D28 and Q34 from TPR1, as well as residues P71 and D72 from TPR2 motif. Figure S6 shows the proximity of these residues to the InvF/ α-CTD complex. Of note, the protein-protein interaction is a dynamic system, i.e., there are various motions and conformational changes occurring under physical conditions. Unfortunately, docking presents only one conformation possible under conditions limited to a rigid system. Thus, it is necessary to evaluate the molecular mechanism of interactions through MD simulations, which will provide valuable insights into structural and functional aspects of protein-protein interactions.

Stabilization of the InvF/ α-CTD complex by SicA

The initial structures for the MD simulations were selected based on the top score of molecular docking between SicA and InvF/ α-CTD complex. Thus, three simulations were performed, evaluating the interaction of SicA through the TPR1 with Cav3, TPR2 with Cav1 and TPR3 with Cav3. The trajectories of 500 ns-MD simulations revealed that all systems are overall stable (Fig. S3) and SicA remains interacting with the complex in the proposed cavities (Fig. S7). Figure 4 shows the distance of key residues for protein-protein interaction.

Figure 4 Interactions between Residues of α-CTD, InvF, and SicA during molecular dynamics simulations.

Distances between the oxygen atom (O) of residue 259 of α-CTD and the nitrogen atom (N) of residue 140 of InvF, as well as between the oxygen atom (O) of residue 294 of α-CTD and the oxygen atom (O) of residue 150 of InvF (A). Distances between the oxygen atom (O) of residue 265 of α-CTD and the nitrogen atom (N) of residue 150 of InvF, as well as between the oxygen atom (O) of residue 294 of α-CTD and the oxygen atom (O) of residue 150 of InvF when SicA was mediated by the TPR1 motif (B), TPR2 motif (C) and TPR3 motif (D). Representation of amino acids, in stick format, used to monitor the distances from InvF to α-CTD for simulations in the absence of SicA (E), with SicA-TPR1 (F), SicA-TPR2 (G) and SicA-TPR3 (H). Simulations were conducted in duplicates to ensure the reliability of the observed interactions.

As we can see in Fig. 4, the simulation replicas (rep1 and rep2) are consistent with each other, showing that the key residues used to monitor stability remain close. This observation reveals the sustained stability and interaction of the complex across both replicas throughout the entire simulation trajectory. However, the binding affinity calculations by PBSA (Table 1) show that there are significant differences in the energy profile of interactions between SicA and the InvF/ α-CTD complex. The SicA interaction occurs preferentially with the motif TPR1 and TPR2, as observed by Li et al. (2022), since they present equivalent energies, −668.14 ± 118.4 and −689.88 ± 111.60 kJ/mol, while the interaction with TPR3 presents energy of −478.44 ± 23.15 kJ/mol. In light of previous data which suggest that SicA forms dimer or higher order oligomers (Darwin, 2001), our results raise the possibility of interactions between the dimeric form of SicA with Cav1 and Cav3, given the similarity of the energy profile and the proximity between them. However, it is not the scope of the present work to evaluate the interaction between the dimeric form of SicA with the InvF/ α-CTD complex, but just reinforce that finding two cavities with similar energy profiles is in agreement with previously reported experimental data.

Table 1 MM-PBSA binding affinity and energy contributions of SicA with the InvF/ α-CTD complex.

Energy contributions (kJ/mol)	TPR1 ± SDa	TPR2 ± SDa	TPR3 ± SDa	
van der Waal energy	−593.15 ± 84.01	−740.06 ± 29.41	−496.76 ± 21.30	
Electrostatic energy	−2,373.52 ± 222.20	−3,133.14 ± 167.20	−2,361.39 ± 114.50	
Polar solvation energy	2,377.75 ± 222.90	3,282.98 ± 158.00	2,447.47 ± 97.47	
SASA energy	−79.22 ± 11.20	−99.66 ± 4.24	−67.77 ± 2.21	
Binding energy	−668.14 ± 118.4	−689.88 ± 111.60	−478.44 ± 23.15	
Note:

a SD means standard deviations between five frames of 2 ns.

SicA appears to be essential for keeping InvF in the ideal conformation, mainly the DNA interaction domain, since the MD simulations showed that in the absence of SicA (Fig. 5A) a conformational change is noted with greater intensity, in the HTH1 ( α6 and α7) and HTH2 ( α9 and α10) motifs, in relation to the simulation data with SicA (Fig. 5B). The structural alignment of the initial and final conformation of the simulation trajectory without SicA produced an RMSD of 4.86 Å, in the 56 residues that form α10 and α7, while in the trajectory containing the structure of SicA, the RMSD value was 2.94Å (Fig. 5D). These conformational changes can also be observed in the RMSF plot (Fig. 5C). High values in the RMSF plot represent variability in the structure of the protein, thus indicating the most relevant amino acids for molecular motion. Thus, structural domains α6, α7, α9 and α10 suffer greater fluctuation in the absence of SicA (black lines) compared to the interaction of SicA through the TPR1 and TPR2 motif, blue and purple line, respectively.

Figure 5 Cartoon representation of first (left) and last (right) frame of MD simulations in the absence (A) and in the presence (B) of SicA. Comparison between the RMSF of InvF backbone in the absence (black line) of SicA and through SicA interactions with TPR1 (blue line), TPR2 (purple line) and TPR3 (brown line) motifs (C). Structural alignment between first and the last frame of InvF obtained from molecular dynamics simulations, in the absence (left) and presence (right) of SicA (D).

Moreover, our results suggest that SicA also stabilizes α-CTD structure in two distinct ways. First, the α-CTD backbone suffers less fluctuation in the positions of atoms in relation to the trajectory without SicA. This suggests that SicA has a role in the backbone stabilization of α-CTD by intermolecular interactions, not allowing the structure to undergo major conformational changes. Second, SicA is able to maintain α-CTD stabilized in a certain region, interacting with α7 and α6 throughout the MD trajectory. On the other hand, in the absence of SicA, the NTD domain becomes free to interact with α-CTD, slightly shifting its structure. Surprisingly, the interaction between NTD and α-CTD increases the binding affinity considerably. Table 2 presents the binding energy between α-CTD and InvF, as well as the polar and non-polar energy contributions.

Table 2 MM-PBSA binding affinity and energy contributions of α-CTD with the InvF protein.

Energy contributions (kJ/mol)	TPR1 ± SDa	TPR2 ± SDa	TPR3 ± SDa	WO SicA ± SDa	
van der Waal energy	−270.99 ± 14.01	−197.50 ± 5.09	−338.44 ± 29.49	−349.08 ± 21.01	
Electrostatic energy	−637.19 ± 143.60	−577.39 ± 54.38	−1,371.69 ± 162.70	−1,751.17 ± 161.90	
Polar solvation energy	681.41 ± 30.12	458.16 ± 85.09	1,148.68 ± 62.04	1,394.24 ± 120.70	
SASA energy	−32.53 ± 1.78	−23.78 ± 1.76	−41.80 ± 2.51	−46.81 ± 3.04	
Binding energy	−259.30 ± 120.50	−340.51 ± 50.81	−603.25 ± 148.50	−752.82 ± 103.30	
Note:

a SD means standard deviations between five frames of 2 ns.

MM-PBSA calculations revealed that SicA does not increase the affinity of α-CTD to the complex, on the contrary, it considerably decreases the binding energy from −752.82 ± 103.30 to −259.30 ± 120.50 and 340.51 ± 50.81 kJ/mol for TPR1 and TPR2 motifs, respectively. In addition, it can be noticed that the binding affinity of α-CTD to InvF, obtained from the trajectory of SicA interacting with the motif TPR3, is closer to the affinity of the trajectory without SicA. However, the affinity of SicA, interacting with the TPR3 motif, is much lower than the affinity when the interaction occurs through the TPR1 and TPR2 motifs (Table 1). These data reinforce that the interaction occurring via TPR3 has little effect on the InvF/ α-CTD complex and points out that the preferential interaction of SicA should occur in cavity 3 and 1 via TPR1 and TPR2, respectively.

A recent study published also by our group (Cortés-Avalos et al., 2024), by using purified versions of InvF and RpoA, with a bacterial two hybrid system, and with the use of RpoA negative dominant mutants, reinforces the predicted interactions between InvF with the RpoA carboxyl domain ( α-CTD). Moreover, the specific interaction between α-CTD with InvF was also experimentally verified. In this regard, interactions of other AraC/XylS regulators with the RNAP in both domains have been shown, such as the NTD in XylS (Ruiz & Ramos, 2001), and the DBD in MelR, SoxS, MarA and Rob (Grainger et al., 2003).

Mapping the main residues responsible for interactions between InvF and α-CTD by computational alanine scanning

Protein-protein interactions present a key role in many biological processes, thus, understanding the molecular mechanism allows the development of relevant strategies for the regulation of several metabolic pathways. For this reason, an alanine scanning approach was conducted to verify the molecular stability of the InvF/ α-CTD complex due the mutations in the InvF amino acids. Based on the PBSA calculations, 12 amino acids (M1, K13, R14, R17, R40, K99, K134, R140, K141, R176, R177 and R181) with values less than -50 kJ/mol were selected for alanine scanning mutagenesis. They represent the greatest contributions to binding affinity. The PBSA data suggest that SicA does not change the interaction profile of the InvF/ α-CTD complex, but a considerable decrease in energy values of some amino acids can be observed (Fig. S8), mainly for residues located in the NTD (K13, R14, K15 and R17) and DBD (R176, R177, R181 and K186) region. On the other hand, only K134, R140 and K141 had a higher energy values with α-CTD in the presence of SicA. Considering that there was no change in the energy profile due to the presence of SicA, we decided to carry out the in silico mutations in the InvF structure, MD simulations with InvF/ α-CTD complex mutaded and PBSA calculations.

The protocol used for alanine scanning mutagenesis was based on PBSA calculations since it is more robust than scoring functions and requires less computational power than thermodynamic integration (TI) method (Wang et al., 2019), being a limiting factor for the evaluation of large numbers of structural perturbations. Furthermore, a study compared the efficiency and accuracy of PBSA against TI demonstrated that PBSA has the same level of accuracy in a fraction of the computational time relative to TI calculations (Martins et al., 2013). Figure 6 presents the results of Alanine-Scanning Mutagenesis (ASM) and its respectively binding energy.

Figure 6 Differences in binding free energies ( ΔΔGbinding) per residues of alanine scanning mutagenesis.

The color scale represents the variation in energy, in KJ/mol, with the greatest variations being represented by the blue color (A). Cartoon representation of the interface between InvF and α-CTD, highlighting residues mutated in sticks (B). Effects of mutation in the binding affinity between InvF and α-CTD (C).

The ASM results showed that the mutations of the selected residues cause a local effect on the structure of the protein, i.e, the mutation decreases the affinity of the mutated amino acid, but does not lead to a disruption of the region or affect the affinity of the other amino acids, so they continue to interact in a very similar way (Fig. 6A). The mutated amino acids are located in the protein-protein interface region (Fig. 6B), except for M1, which was used to assess whether the protocol could introduce any binding energy bias. Therefore, it is supposed that a mutation in these residues can lead to the loss of a specific interaction with the α-CTD residues, causing a decrease in the binding affinity between the proteins.

As expected, the M1A mutation did not produce any significant difference compared to the native protein (2.50 kJ/mol), indicating that the protocol employed did not introduce any artificiality due to the mutation (Fig. 6C). On the other hand, residue mutations located on the HTH motif, such as R176A, R177A and R181A, caused a decrease in relation to wild type binding affinity of 103.90, 165.53 and 162.84 kJ/mol, respectively. These results are consistent with the mutations carried out in the MarA protein, where it was observed that mutations in the residues from the HTH motif decreased remarkably the activity (Gillette, Martin & Rosner, 2000). In addition to the DBD motif, we evaluated the effect of mutations in the NTD region through residues K13A, R14A, R17A, R40A and K99A. Surprisingly, our results revealed an important role of this region for protein-protein interaction, which K13A, R14A, R17A and K99A produced changes, relative to wild type, in binding energy of 102.61, 126.71, 112.23 and 116.73 kJ/mol, respectively. Especially, the alanine substitution R40A decreased 165.34 kJ/mol of binding affinity. Together, these results point to a critical role of the amino acids located in the NTD region for the molecular recognition of α-CTD and agree with the experimental data that showed that mutations in five amino acids (W19, E21, L28, P78 and R110) outside the DBD domains of MarA resulted in severely decreased activity. Furthermore, these residues are highly conserved in the MarA homologs suggesting a common role in overall protein structure or transcriptional activation (Gillette, Martin & Rosner, 2000).

Although the mutations caused a significant decrease in binding energy, it was not enough to observe a disruption in the structure of the InvF/ α-CTD complex, suggesting the relevance of multiple amino acids to binding affinity and a conformational flexibility in the structure that allows protein-protein interaction in many different ways. This phenomenon appears to be similar to experimental data concerning MarA mutations, in which activity was detected in all cases, suggesting that a fraction of the protein remains folded properly or improperly folded protein still able maintain its activity.

In a related study, Kim et al. (2014) observed that point mutations in each of the three TPRs affected their stability when expressed from their own promoter. Moreover, when overexpressed, these mutations were unable to fully activate the expression of sipB and sigD. However, purified versions of these mutants demonstrated the ability to interact with InvF, with TPR2 and TPR3 mutants exhibiting a weaker interaction. Moreover, Li et al. (2022) illustrated the significance of residues within these TPR motifs in sensing c-di-GMP. They observed that a SicA N70A mutant could contact InvF to trigger transcription activation but failed to detect this secondary messenger. These observations shed light on the inability of the TPR mutants reported by Kim et al. (2014) to activate transcription of sipB and sopB despite their interaction with InvF, presumably owing to their effect on the interaction with RpoA.

Conclusions

Studies have pointed to the role of SicA as a cofactor for InvF-dependent transcription activation of the sicA and sigD promoters. However, it is still unclear whether and how they interact. Employing in silico strategies, we focused on identifying potential interaction regions and mechanisms between InvF, SicA, and the α-CTD of RNAP. Our findings demonstrated the successful alignment of the InvF model with the MarA crystal, effectively positioning α-CTD relative to InvF and resulting in a stable protein-protein complex.

Molecular dynamics simulations of the InvF/ α-CTD complex revealed a stable trajectory, accompanied by a noteworthy conformational change in the N-terminus region of InvF. This change facilitated significant intermolecular interactions with α-CTD, suggesting a key role in the complex stability. The exploration of potential cavities in the complex structure identified three regions suitable for interaction with SicA. Systematic molecular docking studies within these cavities highlighted preferential interactions between SicA and the TPR1 and TPR2 motifs, emphasizing their crucial role at the interface of InvF and α-CTD.

Additional molecular dynamics simulations provided insights into the stabilizing function of SicA in the HTH motifs, crucial for DNA interactions. Notably, prior research demonstrated that mutations in the residues of the HTH motif in MarA led to a significant decrease in activity. Although MM-PBSA calculations did not reveal enhanced binding affinity between InvF and α-CTD in the presence of SicA, they suggested a potential role for SicA in preserving the proper conformation, especially within the DNA-binding domain.

To discern the contributions of specific amino acids to protein-protein affinity, we conducted an alanine scanning mutagenesis. Results highlighted the significance of residues R177 and R181 in the DBD, causing substantial changes in binding affinity with α-CTD. Additionally, observations of conformational flexibility in the NTD region emphasized its pivotal role in interacting with interface residues, with a notable decrease in binding energy observed in the R40 mutation. In summary, our study provides valuable insights into the molecular mechanisms governing the interaction between InvF and α-CTD, underscoring the stabilizing influence of chaperone SicA in this complex.

Supplemental Information

Supplemental Information 1 Supporting Information.

Supplemental Information 2 Contribution of energy per residue obtained by MM-PBSA in the absence of SicA.

All the results obtained from molecular dynamics simulations and PBSA.

Supplemental Information 3 Contribution of energy per residue obtained by MM-PBSA in the presence of SicA.

All the results obtained from molecular dynamics simulations and PBSA.

The authors would like to thank Dr. Nina Pastor Colon for her key inputs to the general quality of this manuscript and Dr. María del Carmen Ponce Caballero for providing the necessary infrastructure. The authors would also like to thank Sistema de Computação Petaflópica do SINAPAD for providing the computers for this research.

Additional Information and Declarations

Competing Interests

Author Contributions

Data Availability

The authors declare that they have no competing interests.

André B. Farias conceived and designed the experiments, performed the experiments, analyzed the data, prepared figures and/or tables, authored or reviewed drafts of the article, and approved the final draft.

Daniel Cortés-Avalos analyzed the data, authored or reviewed drafts of the article, and approved the final draft.

J. Antonio Ibarra conceived and designed the experiments, analyzed the data, authored or reviewed drafts of the article, and approved the final draft.

Ernesto Perez-Rueda conceived and designed the experiments, analyzed the data, authored or reviewed drafts of the article, and approved the final draft.

The following information was supplied regarding data availability:

The raw data are available in the Supplemental Files.

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
