# Peer review of "The interaction of InvF-RNAP is mediated by the chaperone SicA in Salmonella sp: an in silico prediction"

_PeerJ, doi:10.7717/peerj.17069_

## Round 0.1 · original submission · Major Revisions

A couple of concerns were raised by the reviewers, some very minor and some major. While you should respond in your revised version to all raised comments either by implementing changes or sufficiently explain why you decided differently, I would like you to focus specifically on few of the major points raised by the reviewer, which are the justification of their usage of GROMOS 54a7 forcefield for MD sims and MM-PBSA calculations, the residues of the homology model of InvF obtained by the Robetta program in the disallowed regions. How do those perform in the template. Why were you not able to obtain a model that does show no residue in the disallowed region and at least 95% residues in the preferred regions? And finally please report the sequence similarity between InvF and the crystal of MarA (PDB ID 1XS9) that were used for the structural alignment, means potentially inclusion of a detailed sequence alignment between MarA and InvF containing information like motifs, s structure, interacting residues.

**Language Note:** The review process has identified that the English language must be improved. PeerJ can provide language editing services - please contact us at copyediting@peerj.com for pricing (be sure to provide your manuscript number and title). Alternatively, you should make your own arrangements to improve the language quality and provide details in your response letter. – PeerJ Staff

Reviewer 1 ·

Basic reporting

The authors present interesting study of interaction between InvF-SicA and RNAP using the computational methods. The study nicely combines several computational approaches such as the protein structure prediction, binding pocket detection, molecular docking, MD simulations, binding energy calculations and alanine scanning mutagenesis to investigate the effect of individual residues on the binding energy.
The manuscript is easily readable written in the understandable English, with exception of the section CONCLUSIONS, where some sentences can be improved. Some examples of the grammatically weird sentences:
Line 335-336: “, but still not clear whether and how could be the mechanism of interactions between them.“
Line 340: „The InvF/α-CTD complex was evaluated by molecular dynamics“ I would add simulations.
Line 346-347: „Molecular dynamics simulations showed the role of SicA in stabilizing the Helix-turn-Helix motifs, important regions for contact with DNA.“

My other comment is concerning discussion of the results of MD simulations. As I have found the Fig S8 very illustrative and nicely demonstrating the stabilizing effect of SciA, I would suggest to transfer the Figure S8 from Supporting Information to the main manuscript.

I would recommend the manuscript for publication after minor revision.

Experimental design

The research questions are well defined. The methods nicely complement each other and they are chosen very well.
My only comment is concerning the selection of the forcefiled. I would just like to ask authors for some justification of their usage of GROMOS 54a7 forcefiled for MD simulations (and I assume also for MM-PBSA calculations) among the other possible forcefileds. I believe that the selected forcfiled is fine for the studied systems, however if one of the future study would also involve simulations of studied protein complexes with DNA (as they potentially could), then Amber forcefiled would be more beneficial.

Validity of the findings

The analysis of the results is robust and the conclusions drawn from the results seem to be correct.

Reviewer 2 ·

Basic reporting

1. The English needs refinement in some places. It would be a good idea to get it reviewed by an English-speaking person. E.g. Line 282: Mapping the main residues ..... instead of mainly residues.

2. The gene names should be italicized throughout the manuscript. E.g. line 53, 335

3. Typhimurium is a serovar and should not be italicized. Please correct it.

4. It would be a good idea to write TTSS as T3SS as this is the most accepted abbreviation these days.

5. It would be great if the authors could simplify sentences 57-59 for clarity.

6. Full-form for TPR should be provided when mentioned for the first time. Line 62: it should be TPR, not TPRs.

7. The section "Binding affinity calculations by MM-PBSA" can be rewritten and the calculations explained for clarity. In the current state, it is confusing for the reader to follow the calculations used.

Experimental design

If the authors can substantiate the data with mutation studies of the important residues and demonstrate the loss of regulation.

Validity of the findings

no comment

Reviewer 3 ·

Basic reporting

no comment

Experimental design

no comment

Validity of the findings

no comment

Additional comments

The submitted manuscript describes an in silico work to understand the interaction between
InvF-SicA and RNAP. There is a need to understand that results obtained using in silico methods
have limited significance till verified using experimentation and it should be reported in the same
approach in any scientific publication.

The structure of InvF used in the study was obtained by the Robetta program which had
93.3% residues in the allowed region and few residues in the disallowed region of
Ramachandran plot. This predicted structure has been used for all the structural analysis. Since
there are residues in the disallowed region; how authors are sure about this structure. Why they
have tried Robetta only for this molecule? They can try to obtain a better model having no
residue in the disallowed region and at least 95% residues in the allowed region.
The predicted structures should also be analyzed on other parameters.

Structural model of the InvF/α-CTD complex was generated using a structural alignment
between InvF and the crystal of MarA (PDB ID 1XS9). Authors have not reported about the
sequence similarity between these two molecules. I would suggest including a detailed sequence
alignment between MarA and InvF containing information like motifs, s structure, interacting
residues ….

Line 186. “it can be seen that some InvF amino acids present in the N-terminal domain (NTD)
have enough flexibility to perform a movement that promotes interactions with amino acids of
the a-CTD (Figure 2B).” How it has been interpreted? Has RMSF been used to verify this?
Alanine scanning mutagenesis has been used to verify the molecular stability. Identification and
pictorial representation of different interactions involving these identified residues would help to
understand the importance of these residues for stability and function in the manuscript.

Figure legend should be more descriptive.

- what is rep1 nd 2 in fig 2. What is NTD …. Figure legend should explain properly…
- Fig 6A legend should be better explained..Explain X and Y axis and significance of color

The language should be clear and typographical errors should be corrected before final
acceptance.

“The gene regulation has as a crucial step …”

Annotated reviews are not available for download in order to protect the identity of reviewers who chose to remain anonymous.

---

## Round 0.2 · Minor Revisions

Reviewer 2 made two suggestions, please address both of them.

Reviewer 2 ·

Basic reporting

The suggestions wrt "Binding affinity calculations by MM-PBSA" have been addressed but the equation (3
) is still not understandable: ∆GbindingEMM = ∆Ebondbonded +∆Enonbond +∆Gpolar +∆Gnonpolar −T∆Snonbonde
Kindly modify.

Experimental design

The authors responded to my comment, " If the authors can substantiate the data with mutation studies of the important residues and demonstrate the loss of regulation."
It is advised that they explicitly state the experimental work exists in the literature to back/support the results of their in-silico analysis.

Validity of the findings

No comments

Additional comments

No comments

---

## Round 0.3 · accepted · Accept

Thank you for submitting your minor revisions. I have assessed your revision myself and can confirm that the revisions address all concerns and comments the reviewers had raised to my satisfaction. Your manuscript now is ready for publication.